# How does the audience return to the music theater? Exploring the influence of musical theater adaptation of Chinese nursery rhymes

Jian Li[1], Yu Hu[2]*

**1** ZhanJiang Preschool Education Collage, Zhanjiang, China, **2** BeiBu Gulf University, Qinzhou, China

* guoshang751103@163.com

## Abstract

Musicals, as an art form, are attracting greater attention in China's increasingly competitive market. In particular, the adaptation of Chinese nursery rhymes in musicals and its influence on audience affective reactions and behavioral intentions has emerged as a notable research focus. This study aims to explore the impact of nursery rhyme adaptations in musicals on audience interaction, affective reactions, satisfaction, and repurchase intention. Using questionnaire surveys and structural equation modeling, we empirically analyse how audience interaction and affective reactions in these musical adaptations enhance satisfaction and increase repurchase intention. The results indicate that audience interaction significantly boosts affective reactions, leading to heightened satisfaction and a greater likelihood of repurchase. Furthermore, elements such as scenic design, lyric adaptation, and melody adaptation enrich the overall artistic experience by influencing both cognitive and affective responses. This study offers strategic recommendations for musical production, specifically highlighting the importance of audience interaction and affective reactions throughout the creative process to maximize engagement and commercial success. In conclusion, our findings provide valuable theoretical and practical insights that not only support cultural inheritance but also drive innovation within the musical theater industry.

## 1 Introduction

The diversification of products supplied by China's performing arts market and the growth of Chinese people's art consumption have made musicals a new growth point in the art market; hence, there is an urgent need to launch a number of new works that can cater to China's consumption demand for musicals, and to break down the linguistic barriers and competitive advantages of traditional musicals. Musicals now stand out in the competition in the performing arts market, and are expected to form an independent industrial chain, further enhancing the internal driving force of the cultural and creative industries and the performing arts market. Correspondingly, the musical industry experienced significant changes and impacts

**Data availability statement:** All relevant data are within the manuscript and its Supporting Information files.

**Funding:** 2021 Featured Innovation Projects of General Colleges and Universities in Guangdong Province 'Research on the dissemination mode of dramatized Leizhou Nursery rhymes under the background of Rural Revitalization Strategy' (2021WTSCX271).

**Competing interests:** The authors have declared that no competing interests exist.

due to external factors such as the pandemic in 2022. For example, Broadway's total box office revenue for the 2022-23 season was 1.578 billion, with an audience of 12.28 million and 1474 weeks of performance. This data has increased compared to the severely affected 2021-22 season by the epidemic, with box office revenue of 845 million and an audience of 6.73 million at that time. In the West End of London, the total box office revenue in 2022 was 892.8 million pounds, with a total of 19224 performances, an increase in revenue and number of performances compared to previous years. These data highlight the gradual recovery trend of the theater industry in response to external disturbances such as the pandemic and theater capacity restrictions. The report also shows that the average ticket price in the Western District has increased to 52.17 pounds.

The nursery rhymes in musicals, as a diverse form of expression that integrates music, drama, and dance, constantly attract social attention and discussion. Also, it creates a mutually dependent and mutually beneficial model between film and literature. Many industry professionals have been striving in recent years to explore and attempt to solve various problems faced in the development of musicals in their respective professional fields, to revitalize the Chinese musicals industry. Some studies focus on the micro level development of musicals, such as script creation, singing methods, and performance themes. As the continuous development of the entertainment industry and the diversification of audience demands, adapting children's rhymes in musicals as a novel and eye-catching art form is gradually becoming a brilliant star on stage. Nursery rhymes play multiple roles in Chinese society, including cultural inheritance, moral education, entertainment and leisure, and social reflection. In recent years, their adaptation in musicals has not only enriched the artistic form, but also injected new vitality into traditional culture. This adaptation reflects the contemporary audience's acceptance and aesthetic needs for traditional culture, and is of great significance for understanding audience preferences, promoting cultural inheritance, and stimulating artistic innovation. It is an important bridge connecting traditional culture and modern aesthetics and promoting the development of cultural undertakings.

The concept of repurchase intention (RI) has been studied extensively over the past decades across both academic and practical fields (Hellier et al. (2003) [1]). Many countries in recent years have carried out nationwide customer satisfaction index evaluations to optimize service levels and enhance industry competitiveness (Pham et al. (2023) [2]). Yet, unlike other entertainment projects, musicals have the function of education and promoting cultural literacy improvement. Cultivate audience loyalty while improving audience satisfaction. Most research on English dramas has focused on music performance issues, with few studies investigating the audience's repurchase intention (Anjani (2021) [3]). Even though some previous studies have attempted to explore the relationship between emotional resonance with the audience through performance techniques in musicals, the aspect of real-time interaction, especially between the audience and the theater, has been overlooked. To bridge the knowledge gap, this research investigates the adaptation of nursery rhymes in musicals, with the aim of utilizing the existing audience and popularity to increase the attention and revenue of musicals. Furthermore, this research also explores various factors in music theaters to enhance the audience's repurchase intention, which not only inherits the emotional and cultural values of the original work, but also showcases unique artistic charm and performance power.

As musical consumption continues to rise, this study investigates audience satisfaction with current adaptations of nursery rhymes in Chinese musicals, focusing on their attitudes and evaluations. An online survey will examine differences in satisfaction based on gender, age, and education, as well as factors influencing audience satisfaction.

Audience attitudes and experiences with nursery rhyme adaptations were analyzed based on feedback and data. Practically, this study can improve the situational sense of artistic presentation and meet original fans 'expectations, though it faces limitations in balancing innovation. The adaptation team should utilize the potential of these works, carefully craft unique scripts, ensure high-quality production through meticulous stage design, captivating music, and talented performances, thereby emphasizing the artistic charm and cultural value of the original work.

In theory, this study provides a new perspective by demonstrating that the adaptation of nursery rhymes in musicals promotes audience repurchase intention, which has received little attention in the literature (Muntean, 2020 [4]; Nijs et al., 2023 [5]). This study further developed a comprehensive mechanism that outlines the interrelationships between various factors, with cognitive and affective reactions acting as mediators and landscape design acting as moderators. Importantly, this study expands the scope of the introduction to clearly describe the specific gaps addressed by the research. We have now explicitly discussed how previous research overlooked the impact of nursery rhyme adaptations in Chinese musicals on audience behavior and return on investment, positioning our work in a unique niche within the academic community.

## 2  Theoretical background

### 2.1 Concept of repurchase intention

The goal of musical marketing is to attract more audiences to enter the theater to watch performances, capture customer interests, attract customer attention, and ultimately enable customers to purchase tickets (Sahin et al. (2011) [6]). Enhancing customer favorability towards a company can effectively increase customer return rates, thereby achieving the goal of repeat purchases by customers (Dandis and Al Haj Eid (2022) [7]). Repurchase Intention (RI) is defined as the customer's willingness or inclination to continue maintaining a trading relationship with existing suppliers (Russo et al. (2017) [8]). Ng et al. (2020). After consuming a product or enjoying a service, customers develop a willingness or tendency to repeat their purchase of the product or service based on their own experience and usage experience [9]). In theory, the concept of customer repurchase intention belongs to the level of customer loyalty. For another, repurchase intention represents a customer's behavioral prediction before product consumption, and is an inherent psychological tendency of the customer, which belongs to behavioral intention. As a psychological state of customers, the relationship between repurchase intention and repeat purchase behavior is the relationship between consciousness and behavior (Alqueza et al. (2022) [10]). Sharma and Singh(2023) have proposed that repurchase intention refers to the tendency or decision of customers to repurchase and consume the same product or service within a certain period after consumption and divides positive repurchase intention into two forms: repurchase and recommended purchase [11]. Sari and Padmantyo (2023)defined repurchase intention as a tendency to maintain a preference for the original supplier during the actual growth of customer demand, which is based on the customer's attitude towards the purchased product or service [12]. Wijarnoko et al. (2023) emphasized that customers' perception and evaluation of the products and services they have consumed determine their subsequent repurchase intention [13].

All in all, the willingness or inclination of customers to repurchase depends on their experience and evaluation of products they have previously consumed. This willingness to repurchase occurs after consumption behavior or experience. This research defines customer repurchase intention as the tendency of customers to choose musical products staged in a certain

theater for consumption again in the context of musical drama consumption. That is, the subjective judgment of the possibility and tendency of customers to purchase specific activities and behaviors such as the theatrical products and services provided by the theater again in the future after having had a theatrical product consumption experience in a certain theater.

## 2.2 Theories of musical adaptation

Bermant(2013) proposed that the adaptation of musicals can leverage their existing popularity and audience to attract more viewers to watch [14]. And it reduces the difficulty of the second creation of the musical script and allows the original work to be disseminated in different artistic forms. Uljasz (2019) pointed out that adapted musicals, as a type of art with a strong sense of ceremony and liveliness, can give the original work a more distinctive style breakthrough and more exaggerated presentation form, highlight the theme and emotions of the original work, and the interaction and improvisation on stage can create more surprises and fun, with a deeper impression and feeling to the audience [15]. However, musicals are a highly comprehensive art form that requires creators and actors to possess various abilities and qualities (Kireyeva (2022) [16]). Many music creations lack unique styles and expressive power. The music in domestic adapted musicals often lacks passion and personality, appearing too plain and monotonous, and failing to effectively integrate with the script content; thus, the adaptation of nursery rhymes in musicals should pay attention to the presentation of story background and character emotions, maintain the artistic and coherent nature of nursery rhymes, bring artistic beauty and emotional resonance to the audience, and thereby increase the audience's purchase rate. Stage play is an art form that has a strong sense of presence and visual impact. High quality stage play can better showcase the scene and atmosphere of the story, satisfy the original fans, achieve a win-win situation in culture and business, and form a brand effect (Hanxi (2023) [17]).

According to Strategy Helix's "China's Performing Arts Market" study, in China, after a significant contraction (−54.9%) in 2020, the revenue of the performing arts market has increased significantly (37.3%), reaching 5.162 billion in 2021. The musical theater market has a very small scale, indicating that the number of domestic musical theater projects is relatively small, and the supply and demand sides of the market do not match, resulting in limited audience choices and skyrocketing ticket prices. Therefore, this study combines the audience's perception of the price of musicals and evaluates both affective and cognitive responses to reflect the symbiotic cycle that exists in actual musical decision-making.

## 3 Hypotheses

### 3.1 Audience satisfaction

Customer satisfaction is an important concept in the field of consumption, which was first proposed by Ge-ling (2006) and has received widespread attention in academia [18]. Defined that customer satisfaction is an emotional response based on a specific product or service transaction, which depends on the degree to which the expected benefits of the product or service are realized. Customer satisfaction is the result of comparing the cost (time, money, effort) of a purchased product or service to the benefits obtained from using the product. In addition to cognitive factors, customer satisfaction is an emotional response (positive and negative) after purchase, which directly affects the customer's evaluation of satisfaction([Meirovich and Bahnan (2008) [19]). Customer satisfaction is the degree to which an individual feels, stemming from their perception of product features or outcomes and their comparison with product expectations (Ladhari et al. (2017) [20]). According to existing

research, audience satisfaction is one of the key factors affecting the willingness of viewers or individuals to continue purchasing. The reason is that when viewers are satisfied with the products or services they purchase or use, they believe that they will benefit more from the related products or services compared to the cost they spend. In other words, in the eyes of the audience, using or purchasing these products or services is worthwhile, thereby enhancing their perceived value and enhancing their willingness to continue purchasing (Utama et al. (2021) [21]). The variable customer satisfaction is replaced with audience satisfaction based on the practical application of this research. Hence, the following hypothesis is constructed.

H1: Audience satisfaction positively correlates with repurchase intention.

## 3.2 Audience interaction

According to the theory of breaking the fourth wall proposed by German playwright Bertolt Brecht, immersive experience can break through the real distance between the audience and the stage, stimulate audience resonance psychology, deepen audience participation, and improve audience immersion in watching. Many music performance works increase the audience's sense of interaction and participation through innovative viewing spaces (Levan (2005) [22]). Immersive music performance art can create a spatial environment that is in line with the background of the play, presenting a unique immersive atmosphere for the audience(Garrido and Macritchie (2020) [23]). Moreover, this atmosphere assists the audience in entering the performance space as soon as possible to enter the plot, laying a foundation for better perception of actor emotions and plot expression (Yan et al. (2021) [24]). In addition, energy transfer is one of the most fundamental abilities of actors in performance. Drama master Michael Chekhov believed that energy is the bridge connecting the body and mind, and the cornerstone of all training and performance. Actors not only need to feel and use energy, but also need to amplify this energy to be felt by the audience below or outside the camera. As for stage actors, they need to amplify their energy as much as possible to influence the audience sitting in dozens of rows or upstairs. There is still controversy over whether an actor's personal energy can truly be amplified to the farthest seat in the theater but using their own energy to convey as many emotions as possible and influence as many audiences as possible is one of the performance pursuits of stage actors. Based on the theoretical arguments and empirical evidence, audience interaction, therefore, included in the research framework as a moderator with the hypotheses formulated as follows.

H2a: Audience interaction moderates the relationship between affective reactions and audience satisfaction.

H2b: Audience interaction moderates the relationship between audience satisfaction and repurchase intention.

## 3.3 Affective reactions

Affective response (AR) can be defined by the dimensions of pleasure, arousal, and dominance (PAD) of emotional responses to external environments(Soleimani and Kobti (2013) [25]). Beatty posit that other emotions, feelings, and emotions (such as liking and intimacy) should also be examined (Detandt et al. (2017) [26]). Previous literature revealed that emotional responses have a significant impact on audience satisfaction (Inan et al. (2022) [27]). In musicals, the audience is influenced by various factors of musical performance, which in turn affects their satisfaction. Hence, the hypothesis is formulated as follows.

H3: Affective reaction positively correlates with audience satisfaction.

### 3.4 Cognitive reactions

Cognitive response (CR) refers to the psychological process that an individual triggers when faced with a stimulus (Lilleker (2014) [28]). Cognitive response is the foundation of emotional response, and information must be considered reliable before establishing emotional attachment to an individual, brand, or product (Balasubramani et al. (2021) [29]). Kimiagari and Malafe (2021) proposed that cognitive evaluation of values was found to have a positive contribution to emotions in terms of satisfaction and can in turn stimulate purchase intention [30]. Hence, the hypothesis is formulated as follows.

H4: Cognitive reactions positively correlates with affective reaction.

### 3.5 Audience expectations

Audience expectations are an important factor affecting the choice of musical performance techniques, which further affects the generation of emotional resonance (Gingras et al. (2016) [31]). Audience expectations for music performance may be influenced by various factors such as culture, education, and personal preferences. Audiences from different cultural backgrounds may have different reactions to specific types of music performance techniques (James(2017) [32]), so performers need to consider the cultural differences of the audience to better meet their expectations. Broughton et al. (2021) suggested that it is necessary to consider audiences from different cultural backgrounds, and therefore may flexibly choose performance techniques to meet the expectations of different audiences and achieve better emotional resonance [33]). The audience's music education and experience can also influence their expectations for music performance. Audiences with higher levels of music education may appreciate complex and highly technical performance techniques more because they can understand and appreciate music more deeply. On the contrary, audiences with lower musical literacy or beginners may prefer simple and easy to understand performance techniques to connect with musicals more easily(Thompson et al. (2023) [34]). Hence, the following hypothesis is constructed.

H5: Audience expectations positively correlates with affective reaction.

### 3.6 Melody adaptation

Adapting melodies is a part of adapting nursery rhymes. In these plots, the music becomes more distinctive in China via incorporating the performance of traditional Chinese instruments such as erhu and pipa, as well as incorporating novel melodic elements. In character portrayal, shaping the character's personality can stimulate the audience's cognitive and emotional responses (Xue (2023) [35]). The musical adheres to the creative concept of originality and localization of music scripts, emphasizing the transmission of historical and cultural information to the audience in the process of adapting the melody (Nalin (2010) [36]). The design of musicals emphasizes the combination of originality and adaptation, cleverly incorporating traditional Chinese music elements into the melody, making the music more distinctive in China (Wu (2023) [37]). Meanwhile, it fully reflects the characteristics of Chinese culture through localization, making it closer to the audience and guiding them to experience the unique charm of local culture in musicals. Hence, the following hypotheses are constructed.

H6: Melody adaptation positively correlates with affective reaction.

H7: Melody adaptation positively correlates with cognitive reactions.

### 3.7 Lyrics adaptation

Similarly, adapting lyrics is a part of adapting nursery rhymes(Cayari (2018) [38]). In recent years, with the popularity of the Internet, popular words on the Internet are more and more popular. Roberts pointed out that during performances, concise and vivid expressions are used to attract the audience's attention, enabling them to better understand and participate (Yoon et al. (2021) [39]). Besides, popular lyrics can better reflect the development of modern society and resonate with the audience (De Mello et al. (2022) [40]). Hence, the following hypotheses are constructed.

H8: Lyrics adaptation positively correlates with affective reactions.

H9: Lyrics adaptation positively correlates with cognitive reactions.

### 3.8 Price perception

Price perception refers to the psychological perception of visitors towards the ticket prices of science and technology museums during or after the visit. Price has always been an important driving factor for purchase intention, as monetary factors when purchasing products and services can affect overall perceived value (Kim et al. (2012) [41]). Viewers will consider the fees paid and the value of the services provided, and then consider their perceived value. ANDARI and Muanas (2023) proposed a consistent conclusion that the perceived value benefits of customers are highly correlated with customer satisfaction and loyalty [42] . They believed that music theaters taking measures to improve audience perceived value will also increase audience satisfaction and repurchase intention. Hence, the following hypothesis is constructed.

H10: Price perception positively correlates with cognitive reactions.

### 3.9 Performer's emotions

Music performance, as an important carrier of music art, not only provides a platform for musicians to showcase their skills, but also serves as a bridge for emotional communication between the audience and music. The emotional resonance experienced by the audience in music performances often intoxicates them and makes them feel the profound meaning of life. Performers can form emotional highs and lows in music through clever dynamic processing of music. By highlighting changes in intensity and volume, performers can make the music become passionate and exciting at a certain moment, triggering excitement and passion from the audience (Yue (2021) [43]). The ups and downs of emotions and the dynamic handling often make the audience feel the tension and changes of emotions in music, further deepening emotional resonance (Eerola et al. (2022) [44]). Emotional resonance is not only limited to the subjective experience level of emotions, but also often accompanied by changes in physiological reactions. The physiological reactions of the audience, such as an increase in heart rate and changes in skin conductivity, are closely related to the performer, indicating the existence of emotional resonance (Chang et al. (2022) [45]). Therefore, the following hypothesis is proposed.

H11: Performer's emotions positively correlate with affective reaction.

### 3.10 Scene design

Scene design, as a core element of the environmental space composition of musical works (Naversen (2011) [46]), carries the important mission of shaping the overall style and atmosphere of the work. An excellent scene design not only closely connects with the plot, providing a solid foundation for character shaping and plot development, but also leads the audience

to immerse themselves and feel the deep emotions and artistic conception that the work aims to convey (Hashim et al. (2023) [47]). The scene design should correspond to the theme of the musical. The realism and immersion brought by this atmosphere can make the audience feel as if they are immersed in the world depicted in the work, resonating deeply with history, nature, and characters. In addition, the theater can adopt virtual scenes, which can not only break through the limitations of reality, but also bring a brand-new visual experience to the audience (Ahmed Hussein Hassan (2020) [48]). The use of color and light and shadow is the finishing touch in scene design. Color and light play a crucial role in creating a scene atmosphere and conveying emotions (Lei et al. (2021) [49]). Moreover, the relationship between props and characters is also an important aspect that needs to be considered in Taoist musicals. By equipping characters with appropriate props, their personality traits and status can be highlighted, enhancing their image shaping. Hence, the hypothesis is formulated as follows.

H12: Scene design moderates the relationship between performer's emotions and affective reaction.

## 3.11 Conceptual model

The conceptual model is shown as Fig 1.

# 4 Experimental results

## 4.1 Source of questionnaire questions

This research questionnaire is divided into two sections, aiming to explore the impact of musical adaptations of nursery rhymes on audience behavior. The first section covers emotional and cognitive factors (performers' emotions, cognitive and affective reactions, and audience expectations) as well as performance content and design factors (lyric adaptation, melody adaptation, and scenic design) to assess the impact of artistic expression on audience experience. The second section includes situational questions focusing on audience interaction, satisfaction, and repurchase intentions, measuring audience engagement and behavioral intentions. To ensure validity, the measurement items were sourced from previous academic

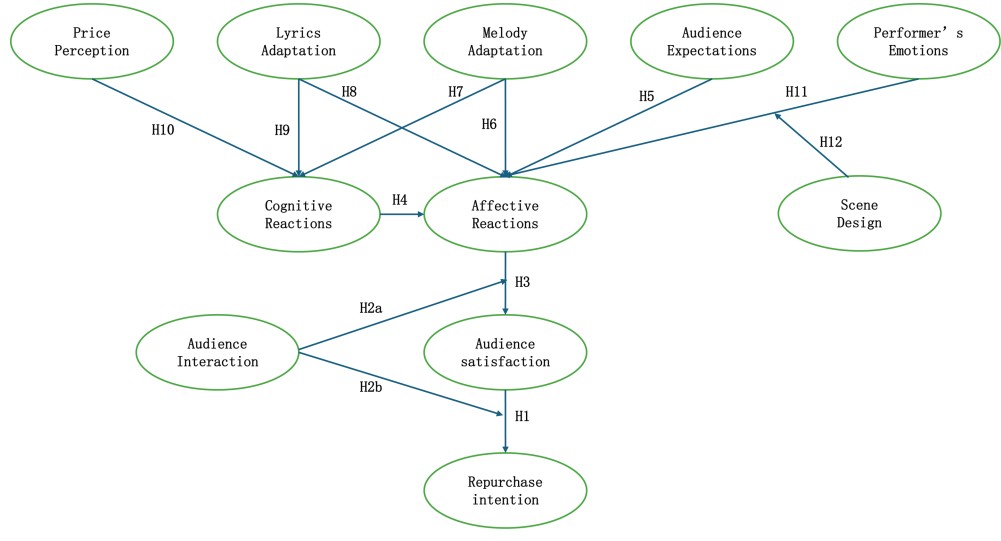

**Fig 1. Research model.**

studies (see Table 1), ensuring both the scientific rigor and practical relevance of the research. The questionnaire is shown in Table 12 in the Appendix.

The recruitment period for this study spanned from May 15 to July 15, 2024, following nursery rhyme adaptation performances at two music theaters in Guangzhou, China. To collect raw data, a self-administered online questionnaire survey was conducted. The research subjects are audience members who have come to the theater to watch performances to ensure the representativeness of the sample. A total of 311 valid and qualified responses were collected.

## 4.2 Participant demographics

The participants of the questionnaire were audience members who had watched performances at two theaters in Guangzhou, and a total of 326 responses were received. After excluding 15 responses that did not meet the inclusion criteria, 311 valid responses were retained for analysis. This study used a random sampling method to investigate these audiences (the results are shown in Table 2).

## 4.3 Factor analysis

**4.3.1 Reliability and validity analysis** Cloning the Bach Alpha value to evaluate the internal consistency of a questionnaire or scale, i.e. whether each item measures the same concept in a coordinated and consistent manner. A Cronbach alpha coefficient higher than 0.7 is generally considered to have good consistency. In Table 2, the Clonbach Alpha value is 0.911, indicating a very high internal consistency of the questionnaire, which helps to ensure the reliability of the research results. The Kaiser Meyer Olkin (KMO) test is used to measure whether the data is suitable for factor analysis. The closer the value is to 1, the more suitable the data is for factor analysis. The suitability score for KMO sampling is 0.859, which is much higher than the baseline value of 0.6, indicating that the data is very suitable for factor analysis. The Bartlett's sphericity test is used to test whether there is a correlation between variables. A p-value less than 0.05 usually indicates sufficient correlation between variables and can be used for factor analysis. In this study, the p-value of the Bartlett sphericity test was 0.000, further confirming the significant correlation between variables, as shown in Table 3.

**4.3.2 Factor analysis** As shown in Table 4, this research extracted 11 factors from the data, and the initial eigenvalues of each factor were greater than 1, indicating that each factor can explain a certain amount of variation in the data. After factor rotation, we obtained

**Table 1. Sources of measurement items in the questionnaire.**

| Constructs | Items | Sources |
|---|---|---|
| Price Perception | 3 | Kim et al. (2012) [41]), Raja et al. (2020) [50] |
| Lyrics Adaptation | 3 | Cayari (2018) [38], De Mello et al. (2022) [40] |
| Melody Adaptation | 3 | Xue (2023) [35], Wu (2023) [51] |
| Audience Expectations | 4 | Kobti et al. (2013) [52], Broughton et al. (2021) [33] |
| Performer's Emotions | 4 | Juslin et al. (1996) [53], Eerola et al. (2022) [44] |
| Cognitive Reactions | 4 | Broughton et al. (2021) [33], Japutra et al. (2023) [54] |
| Affective Reactions | 3 | Nazarov et al. (2020) [55], Hashim et al. (2023) [47] |
| Scene Design | 3 | Brockett et al. (2010) [56], Lei et al. (2021) [49] |
| Audience Interaction | 4 | James (2017) [32], Yan et al. (2021) [24] |
| Audience Satisfaction | 3 | Soleimani and Kobti (2013) [25]), Inan et al. (2022) [27] |
| Repurchase Intention | 3 | Zhao et al. (2012) [57], Sari et al. [58] |

**Table 2. Profile of respondents (n=311).**

| Demographics | Frequency | Percentage (%) |
|---|---|---|
| **Gender** | | |
| Male | 191 | 61.4 |
| Female | 120 | 38.6 |
| **Age** | | |
| Under 18 years old | 27 | 8.7 |
| Age 19-30 | 61 | 19.6 |
| 31-45 years old | 87 | 27.9 |
| Age 45 and above | 136 | 43.8 |
| **Current Education Level** | | |
| Secondary school diploma | 20 | 6.4 |
| Associate Degree | 69 | 22.1 |
| Bachelor's degree | 131 | 42.1 |
| Graduate or above | 91 | 29.3 |
| **How many times have you been to the theater?** | | |
| 1-3 times | 36 | 11.6 |
| 4-6 times | 155 | 49.8 |
| 7-10 times | 88 | 28.3 |
| 10 and above | 32 | 10.3 |
| **Types of Programs** | | |
| Business | 60 | 19.3 |
| Engineering | 23 | 7.4 |
| Information Technology | 16 | 5.1 |
| Mass Communication | 80 | 25.7 |
| Pre-University | 69 | 22.2 |
| Others | 63 | 20.3 |
| **Total** | 311 | 100 |

**Table 3. Reliability and validity analysis.**

| Item | Alpha | Number of Items | KMO Value | Approx. Chi-Square | df | p-value |
|---|---|---|---|---|---|---|
| Value | .911 | 37 | .859 | 5488.732 | 666 | .000 |

**Table 4. Factor analysis results.**

| Factor | Before Rotation | | | After Rotation | | |
|---|---|---|---|---|---|---|
| | Eigenvalue | Variance Explained % | Cumulative % | Eigenvalue | Variance Explained % | Cumulative % |
| 1 | 9.035 | 24.419 | 24.419 | 2.897 | 7.831 | 7.831 |
| 2 | 2.658 | 7.185 | 31.604 | 2.825 | 7.636 | 15.467 |
| 3 | 2.363 | 6.386 | 37.990 | 2.798 | 7.562 | 23.029 |
| 4 | 2.229 | 6.025 | 44.015 | 2.754 | 7.443 | 30.472 |
| 5 | 1.969 | 5.321 | 49.336 | 2.357 | 6.371 | 36.844 |
| 6 | 1.789 | 4.835 | 54.171 | 2.326 | 6.285 | 43.129 |
| 7 | 1.773 | 4.793 | 58.964 | 2.310 | 6.244 | 49.373 |
| 8 | 1.542 | 4.166 | 63.130 | 2.304 | 6.227 | 55.600 |
| 9 | 1.345 | 3.636 | 66.766 | 2.197 | 5.937 | 61.537 |
| 10 | 1.192 | 3.221 | 69.986 | 2.174 | 5.876 | 67.413 |
| 11 | 1.074 | 2.902 | 72.888 | 2.026 | 5.478 | 72.888 |

a new proportion of each factor's explanation of the data variance. The sum of squared loads after rotation indicates that these factors account for a total of 67.413% of the data variance, effectively demonstrating the explanatory power of our experiment and helping to better understand the contribution of each factor to the overall data variability.

## 4.4 Confirmatory analysis

**4.4.1 Model indicator analysis**  This research uses confirmatory factor analysis (CFA) to test the fit of the model. Table 5 shows that the CMIN/DF value is 1.598, which is significantly lower than 3, while the RMSEA value is 0.044, which is also much lower than 0.08, indicating that the model has a good fit. Although NFI, RFI, and GFI were slightly lower than the ideal value of 0.9, the high values of IFI, TLI, and CFI compensated for this difference, indicating that the model adapted well to the analysis variables and effectively supported the construction of the model and demonstrated the effectiveness of the questionnaire. Through these fitting indicators, we can confirm that the model is statistically effective and suitable for further in-depth exploration and validation of theoretical hypotheses.

**4.4.2 Convergence validity analysis**  To evaluate the aggregated validity of variables in the model, this research conducted mean extracted variance (AVE) and composite reliability (CR) indicators. As shown in Table 6, the standardized factor loadings ranged from 0.722 to 0.832, and all significant p-values were less than 0.05, indicating a significant correlation between the latent variable and the observed variable. The AVE values range from 0.747 to 0.804, and the CR values range from 0.820 to 0.875, all of which meet the good criteria for aggregated validity (AVE>0.5, CR>0.7), indicating a high degree of consistency and reliability among the variables in the model.

## 4.5 Structural equation model path analysis

Based on the above results, there are nine potential variables in the structural equation model. Before conducting path analysis, this study confirmed that the model fit well and further analysed the model path coefficients to verify the assumed relationship:

**4.5.1 Structural equation model path analysis.**  As shown in Table 7, cognitive reactions are significantly positively influenced by price perception and lyrics adaptation, while melody adaptation has no significant impact on cognitive reactions. Affective reactions is significantly positively influenced by melody adaptation and audience expectations, while lyrics adaptation and performer's emotions have no significant impact on affective reactions. In addition, Audience satisfaction is significantly influenced by affective reaction, while Recurrence intention is significantly influenced by audience satisfaction. The research explained the dynamic relationships and interactions between different latent variables in the model, and the questionnaire supported the hypothesis. However, it also found that the influence of some pathways was not significant, and the questionnaire did not support the hypothesis.

**4.5.2 Scene design is situated between the impact of performer's emotion on affective reactions.**  Model 1 in Table 8 demonstrates the direct impact of scene design on performer's

**Table 5. Model fit of validation factor analysis.**

| Model Fit | CMIN | DF | CMIN/DF | NFI | RFI | IFI | TLI | CFI | GFI | RMSEA |
|---|---|---|---|---|---|---|---|---|---|---|
| Fit Results | 615.242 | 385 | 1.598 | 0.863 | 0.845 | 0.944 | 0.936 | 0.943 | 0.891 | 0.044 |
| Judgment Std. | | | <3 | >0.9 | >0.9 | >0.9 | >0.9 | >0.9 | >0.9 | <0.08 |

**Table 6. Convergence validity analysis results.**

| Variable | Price Perception | Lyrics Adaptation | Melody Adaptation | Audience Expectations | Performer's Emotions | Affective Reactions | Audience Interaction | Audience satisfaction | Repurchase intention |
|---|---|---|---|---|---|---|---|---|---|
| AVE | 0.785 | 0.804 | 0.797 | 0.760 | 0.755 | 0.747 | 0.768 | 0.789 | 0.776 |
| CR | 0.860 | 0.875 | 0.860 | 0.840 | 0.830 | 0.820 | 0.850 | 0.855 | 0.845 |

**Table 7. Structural equation model path coefficient test.**

| Path | Unstd. Coef. | Std. Error | C.R. | P-Value | Std. Coef. | Conclusion |
|---|---|---|---|---|---|---|
| Cognitive Reactions ← Price Perception | 0.230 | 0.062 | 3.721 | 0.001*** | 0.265 | Supported |
| Cognitive Reactions ← Lyrics Adaptation | 0.211 | 0.060 | 3.511 | 0.001*** | 0.238 | Supported |
| Cognitive Reactions ← Melody Adaptation | 0.074 | 0.058 | 1.277 | 0.201 | 0.088 | Not Supported |
| Affective Reactions ← Lyrics Adaptation | 0.036 | 0.061 | 0.587 | 0.557 | 0.038 | Not Supported |
| Affective Reactions ← Melody Adaptation | 0.234 | 0.064 | 3.629 | 0.001*** | 0.263 | Supported |
| Affective Reactions ← Audience Expectations | 0.203 | 0.069 | 2.947 | 0.003** | 0.209 | Supported |
| Affective Reactions ← Performer's Emotions | 0.125 | 0.068 | 1.844 | 0.065 | 0.125 | Not Supported |
| Affective Reactions ← Cognitive Reactions | 0.280 | 0.071 | 3.938 | 0.001*** | 0.265 | Supported |
| Audience Satisfaction ← Affective Reactions | 0.416 | 0.076 | 5.479 | 0.001*** | 0.387 | Supported |
| Repurchase Intention ← Audience Satisfaction | 0.384 | 0.070 | 5.527 | 0.001*** | 0.390 | Supported |

* p<0.05; ** p<0.01; *** p<0.001.

**Table 8. The mediating role of performer emotion and audience expectations in the influence of scene design.**

| | Model 1 | Model 2 | Model 3 |
|---|---|---|---|
| Constant Term | 2.100** (44.781) | 2.100** (45.227) | 2.065** (43.356) |
| Scene Design | 0.296** (6.404) | 0.262** (5.530) | 0.238** (5.003) |
| Performer Emotion | | 0.142** (2.695) | 0.108** (2.028) |
| Scene Design * Performer Emotion | | | 0.143** (2.783) |
| Sample Size | 311 | 311 | 311 |
| $R^2$ | 0.116 | 0.136 | 0.157 |
| Adjusted $R^2$ | 0.113 | 0.130 | 0.149 |
| F Value | F (1, 313) = 41.014, p<0.000 | F (2, 312) = 24.549, p<0.000 | F (3, 311) = 19.302, p<0.000 |
| Delta $R^2$ | 0.116 | 0.020 | 0.021 |
| Delta F Value | F (1, 313) = 41.014, p<0.000 | F (1, 312) = 7.263, p = 0.007 | F (1, 311) = 7.746, p = 0.006 |

*$p$<0.05; **$p$<0.01. The value of $t$ is enclosed in parentheses.

emotion (significant p<0.01), confirming that scene design has a significant positive impact on performer's emotion. Model 2 introduces affective reactions as a mediating variable, showing a significant impact (p<0.01) on performer's emotions, suggesting that affective reactions play a mediating role in the relationship between scene design and performer emotions. Model 3 further illustrates the interaction term between scene design and affective reactions, which is significant (p<0.01), indicating a significant mediating effect of affective reactions e. That is, the impact of scene design on affective reactions varies with different levels of affective reactions. These results reveal a significant impact of scene design, performer emotions, and their interactions on affective reactions, particularly the significant regulatory role of affective reactions in the effect of scene design on performer emotions.

**4.5.3 Audience Interaction mediates the effect of affective reactions on Audience satisfaction.** Model 1 in Table 9 shows the direct impact of affective reactions on audience satisfaction, with a statistical significance coefficient of 0.142 * * (2.695), indicating a positive and significant relationship between the two. Model 2 introduces audience expectations as an additional variable and reveals a significant impact on emotional feedback and audience satisfaction, with a coefficient of 0.296 * (6.404). In addition, audience satisfaction still holds significant importance, although the coefficient has slightly decreased to 0.108 (2.028). These findings suggest that affective reactions and audience satisfaction play a crucial role

**Table 9. The mediating role of affective reactions and satisfaction in the impact of audience interaction.**

| | Model 1 | Model 2 | Model 3 |
|---|---|---|---|
| Constant Term | 2.100 (44.781) | 2.100 (45.227) | 2.065 (43.356) |
| Affective Reactions | 0.296 (6.404) | 0.262 (5.530) | 0.238 (5.003) |
| Audience Satisfaction | | 0.142 (2.695) | 0.108 (2.028) |
| Affective Reactions * | | | |
| Audience Satisfaction | | | 0.143 (2.783) |
| Sample Size | 311 | 311 | 311 |
| $R^2$ | 0.021 | 0.046 | 0.067 |
| Adjusted $R^2$ | 0.018 | 0.040 | 0.060 |
| F Value | F (1, 313) = 6.699, $p$ = 0.010 | F (2, 312) = 7.164, $p$ = 0.001 | F (3, 311) = 4.827, $p$ = 0.003 |
| Delta $R^2$ | 0.021 | 0.025 | 0.021 |
| Delta F Value | F (1, 313) = 6.699, $p$ = 0.010 | F (1, 312) = 7.164, $p$ = 0.001 | F (1, 311) = 4.827, $p$ = 0.003 |

* $p<0.05$; ** $p<0.01$. The value of $t$ is enclosed in parentheses.

in shaping the impact of audience interaction. In addition, Model 3 is the most comprehensive among the three, including audience affective reactions and satisfaction, as well as interaction. The adjusted R $^2$ value increased from 0.018 in Model 1 to 0.060 in Model 3, indicating an improvement in the model's fit. The significant coefficients of affective reactions (0.238 * [5.003]) and satisfaction (0.143 * [2.783]) in Model 3 indicate that these two variables are influenced by audience interaction. Therefore, audience interaction can liven up the atmosphere on site, providing a better viewing experience and increasing satisfaction.

**4.5.4 Audience interaction mediates the effect of audience satisfaction on repurchase intention.** Model 1 in Table 10 demonstrates the direct effect of audience satisfaction on Audience Interaction (Significance p<0.01), confirming that audience satisfaction has a significant positive effect on audience interaction. Model 2 introduces repurchase intention as a mediator variable and shows a significant effect of repurchase intention on audience interaction (significant p<0.01), implying the role of repurchase intention in the audience satisfaction and audience interaction as a mediator in the relationship between audience satisfaction and audience interaction. Model 3 was further added to account for the interaction term between audience satisfaction and repurchase intention, which was significant (p<0.01), suggesting a significant mediating role of repurchase intention, i.e., the effect of AS on audience interaction varied with the different levels of repurchase intention. These results reveal the significant effects of audience satisfaction, repurchase intention and their interactions on audience interaction, especially the significant moderating role of RI in the effect of audience satisfaction on audience interaction. significant moderating role of RI in the effect of audience satisfaction on audience interaction.

## 4.6 Discussion

Through factor analysis, confirmatory analysis, moderation effect analysis, and path analysis of structural equation modeling, this study validated the hypothesis results and path coefficients shown in Table 11 and Fig 2. The study unveiled intricate relationships among different variables, particularly emphasizing the pivotal roles of audience interaction, affective reactions, audience satisfaction, and repurchase intention. The key findings include:

There is a significant positive correlation (H1) between audience satisfaction and repurchase intention, reinforcing the notion that audience satisfaction has a substantial impact on repurchase behavior. Audience interaction significantly moderates the relationship between

**Table 10. The mediating role of audience satisfaction and repurchase intention in audience interaction.**

|  | Model 1 | Model 2 | Model 3 |
|---|---|---|---|
| Constant Term | 2.100 (44.781) | 2.100 (45.227) | 2.065 (43.356) |
| Audience Satisfaction | 0.296 (6.404) | 0.262 (5.530) | 0.238 (5.003) |
| Repurchase Intention |  | 0.142 (2.695) | 0.108 (2.028) |
| Audience Satisfaction * |  |  |  |
| Repurchase Intention |  |  | 0.143 (2.783) |
| Sample Size | 311 | 311 | 311 |
| $R^2$ | 0.035 | 0.062 | 0.083 |
| Adjusted $R^2$ | 0.032 | 0.056 | 0.076 |
| F Value | F (1, 313) = 11.212, $p$ = 0.001 | F (2, 312) = 10.134, $p<0.001$ | F (3, 311) = 6.739, $p<0.001$ |
| Delta $R^2$ | 0.035 | 0.027 | 0.021 |
| Delta F Value | F (1, 313) = 11.212, $p$ = 0.001 | F (1, 312) = 10.134, $p<0.001$ | F (1, 311) = 6.739, $p<0.001$ |

$p<0.05$; ** $p<0.01$. The value of $t$ is enclosed in parentheses.

**Table 11. Hypothesis classification.**

| No. | Hypothesis | Result |
|---|---|---|
| H1 | Audience satisfaction positively correlates with repurchase intention. | Supported |
| H2a | Audience interaction moderates the relationship between affective reactions and audience satisfaction. | Supported |
| H2b | Audience interaction moderates the relationship between audience satisfaction and repurchase intention. | Supported |
| H3 | Affective reactions positively correlates with audience satisfaction. | Supported |
| H4 | Cognitive reactions positively correlates with affective reactions. | Supported |
| H5 | Audience expectations positively correlates with affective reactions. | Supported |
| H6 | Melody adaptation positively correlates with affective reactions. | Supported |
| H7 | Melody adaptation positively correlates with cognitive reactions. | Not Supported |
| H8 | Lyrics adaptation positively correlates with affective reactions. | Not Supported |
| H9 | Lyrics adaptation positively correlates with cognitive reactions. | Supported |
| H10 | Price perception positively correlates with cognitive reactions. | Supported |
| H11 | Performer's emotions positively correlates with affective reactions. | Not Supported |
| H12 | Scene design moderates the relationship between performer's emotions and affective reactions. | Supported |

affective reactions and audience satisfaction (H2a), as well as the relationship between audience satisfaction and repurchase intention (H2b). This underscores the crucial role of audience interaction in amplifying affective reactions and facilitating the conversion of satisfaction into repurchase behavior. A significant positive correlation (H3) exists between affective reactions and audience satisfaction, indicating that affective reactions are a vital factor in enhancing audience satisfaction.

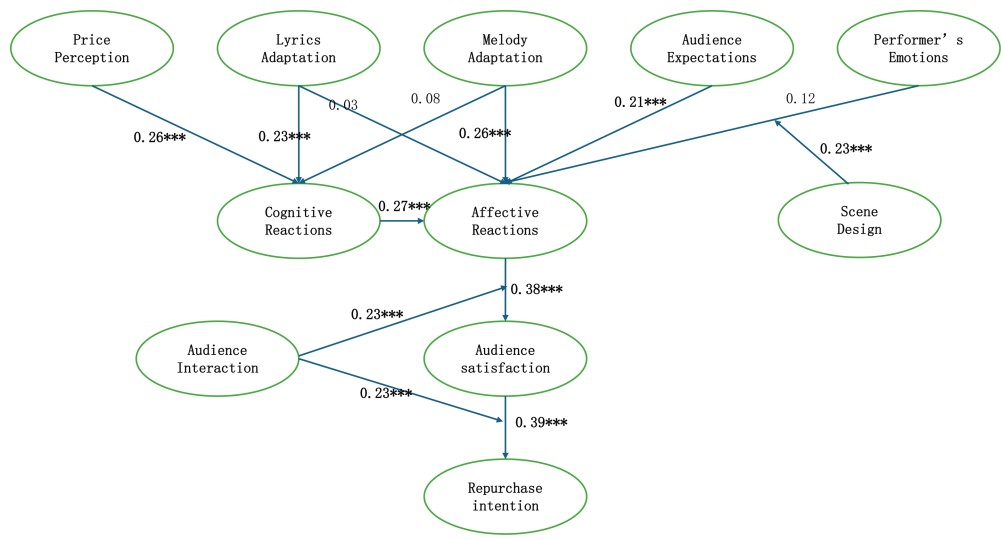

**Fig 2. Assuming path coefficients.**

Cognitive reactions is positively correlated with affective reactions (H4), and the adaptability of song melody (H6) also shows a positive correlation with affective reactions. This suggests a close interplay between music content and affective reactions. While the impact of lyrics adaptability on cognitive reactions (H9) and affective reactions (H8) was not significant, it nevertheless offers a fresh perspective. Notably, a positive correlation was observed between price perception and cognitive reactions (H10), providing an interesting avenue for future research.

Performer emotions have a significant positive impact on affective reactions (H11), highlighting the importance of performer emotions in evoking emotional reactions from the audience.

The moderating effect of scene design on the relationship between performer's emotions and affective reactions has been confirmed (H12). This indicates that appropriate scene design can significantly enhance the communication of performer's emotions and the audience's affective reactions. Regarding the non-significant paths, particularly the lack of significant impact of lyrics content depth and artistry on affective reactions, several potential reasons can be explored. It may be that the selected nursery rhymes, despite their cultural significance, did not resonate deeply with the audience in terms of emotional content or artistic expression. Additionally, the adaptation process may have altered the original essence of the rhymes in a way that weakened their emotional impact. Future research could delve deeper into these aspects, exploring how lyrics content depth and artistry can be better integrated into musical theater adaptations to evoke stronger affective reactions from the audience.

Overall, this research underscores the importance of audience interaction and scene design in enhancing affective reactions and satisfaction. For future research, it is imperative to explore strategies for strengthening the connection between audiences and music theaters through educational and cultural interventions, particularly in the context of adapting Chinese nursery rhymes into musical works. By optimizing these elements, we can not only enrich the performing arts of music theaters but also enhance the overall audience experience, thereby fostering the inheritance and innovation of cultural heritage.

## 5 Implications

### 5.1 Theoretical implications

This research deepens the understanding of the interrelationships between audience interaction, affective reactions, audience satisfaction, and repurchase intention in musicals, especially in the context of children's rhyme adaptation in musicals. Audience interaction significantly moderates the relationship between affective reactions and satisfaction, and further influences repurchase intention. This finding emphasizes the core role of audience interaction in enhancing experience and converting satisfaction into loyalty, consistent with the research findings (Endo and Yoshioka (2021) [59]), which also confirms that audience interaction enhances the attractiveness and retention rate of artistic experiences. The positive impact of affective reactions on audience satisfaction highlights the crucial role of emotional response in improving satisfaction and loyalty, which is consistent(E. and Oh (2022) [60]), who found that emotional resonance significantly increases audience emotional engagement and satisfaction. Meanwhile, the positive correlation between cognitive response and affective reactions further reveals how audiences respond to musical content from both cognitive and emotional perspectives, supporting Wang et al. (2023)view on the interaction between cognition and emotion in artistic experience [61] . Explored the impact of scene design on the moderating effect between performer emotions and audience affective reactionss and confirmed that appropriate scene design can significantly enhance the communication of performer emotions and audience affective reactions. This not only expands the boundaries of musical theory research, but also provides valuable empirical support for the production and performance practice of musicals, emphasizing the importance of considering audience interaction, affective reactions, and scene design in the production process to optimize audience experience and promote cultural inheritance and innovation.

### 5.2 Practical implications

The practical strategy of adapting Chinese nursery rhymes into musicals emphasizes the importance of considering the diverse backgrounds, cultural preferences, and viewing styles of the audience. The production team can enhance audience engagement and satisfaction by designing interactive activities that align with their cultural and affective reactions. This is not only applicable to large theaters, but also to audiences with different economic conditions, ensuring that everyone can enjoy the charm of musicals. Although high-end stage equipment can provide abundant performance resources, the improvement of audience satisfaction depends more on personal investment, cultural literacy, and production quality. In addition, according to (Wu (2022) [62]), effective au- dience interaction strategies significantly enhance emotional engagement and satisfaction; Tronstad, Tronstad (2022) pointed out that modern adaptation can attract more young audiences; Omar [63], Wang et al. (2023) emphasizes that high-quality visual and auditory elements can enhance the audience experience [64].

 Moreover, in areas with rich cultural heritage, incorporating local cultural elements into music adaptations by staff may resonate more deeply with the audience. Meanwhile, innovative solutions such as community-based production or digital platforms can be explored to make musicals more accessible in economies with limited resources. Based on these studies, we provide a series of evidence-based practical recommendations to promote the development of the musical industry and optimize the audience experience.

## 6 Limitations and future research

There are some important caveats to the research that deserve attention. First, the samples used in this study were targeted at the surveyed audience, which may limit the applicability of the results to a wider population. Future research could benefit from expanding the geographical and demographic scope of the sample to validate the research findings and ensure their robustness across different audiences. Next, the study also focused on specific elements such as landscape design, lyrics adaptation, and melody adaptation. Although these factors are crucial, there may be other factors that affect audience engagement and satisfaction in musical adaptations, which were not explored in this study. Future research can delve deeper into other potential variables, such as director selection, actor performance, or cultural background, to gain a more comprehensive understanding of the driving factors behind audience responses. Future research can also incorporate cross-cultural comparisons, rather than being limited to one's own culture to promote cultural exchange between countries. Third, the methods used in this study rely on questionnaire surveys and structural equation modeling. Although these methods are highly suitable for exploring the relationships between variables of interest, they do not provide in-depth qualitative insights into the audience's experience and perception. Combining qualitative research methods such as interviews or focus groups can provide richer data and a deeper understanding of audience reactions to musical adaptations. Finally, future research should address these limitations by examining a wider and more diverse sample, considering other influencing factors, and combining qualitative research methods to gain a more comprehensive understanding of audience reactions in the musical industry.

## 7 Conclusion

This study empirically analyzed how musical adaptations of nursery rhymes influence audience interaction, affective reactions, satisfaction, and repurchase intentions. Factor analysis, validation analysis, and structural equation modeling identified audience interaction and affective reactions as key contributors to satisfaction and repurchase intention. Specifically, the direct positive impact of audience interaction on affective reactions highlights the importance of engaging and immersive theatrical experiences. The results revealed that audience interaction directly enhanced affective reactions, which significantly boosted satisfaction and, consequently, repurchase intention. In the context of nursery rhyme adaptations, the emotional richness and cultural resonance of the content facilitated emotional connections, making audiences more likely to return. Although lyrical adaptation had no direct effect on affective reactions, it indirectly enhanced cognitive responses by improving price perception, thereby deepening the overall audience experience. Scenic design significantly enhanced performers 'emotional expression and the audience's affective reactions, further increasing satisfaction and likelihood of returning. The theoretical significance of this study is that it helps to understand how musical adaptations effectively attract audiences through emotional and interactive elements. In fact, our research findings provide insights for musical producers and directors on how to create adaptations that resonate with audiences, cultivate loyalty, and repeat visits. Future research could provide additional avenues for investigating the role of other factors, such as social media engagement or word-of-mouth marketing, in promoting musical attendance. At the same time, it can provide a deeper understanding of the sustained impact of musical adaptations on audience engagement.

## Appendix

**Table 12. Questionnaire.**

| S/no | Construct | Items |
|------|-----------|-------|
| 1 | Price Perception | I think the price of the musical ticket is reasonable. |
| 2 | | I think the price I paid matches the value I received. |
| 3 | | I think the price-performance ratio of this musical is higher than others. |
| 1 | Lyrics Adaptation | I can understand the adaptation of nursery rhyme lyrics in the musical. |
| 2 | | I think the adaptation of the nursery rhyme lyrics is innovative. |
| 3 | | I find the adaptation of the nursery rhyme lyrics appealing. |
| 1 | Melody Adaptation | I can recognize the melody of the nursery rhyme. |
| 2 | | I think the adaptation of the nursery rhyme melody evokes emotional resonance. |
| 3 | | I think the adaptation of the nursery rhyme melody is innovative. |
| 1 | Audience Expectations | I have high expectations for the content adaptation in the musical. |
| 2 | | I have high expectations for the performance of the actors in the musical. |
| 3 | | I have high expectations for the stage design of the musical. |
| 4 | | I hope to have a good affective reactions from the musical. |
| 1 | Performer's Emotions | I think the performers effectively convey the emotions of the characters. |
| 2 | | I feel the performers' emotional expressions are genuine and natural. |
| 3 | | I can immerse myself in the emotional expressions of the performers. |
| 4 | | I really like the way the performers express emotions. |
| 1 | Cognitive Reactions | I understand the theme and storyline of the musical. |
| 2 | | I can maintain my focus while watching the musical. |
| 3 | | I pay close attention to the adaptation of the nursery rhymes in the musical. |
| 4 | | I pay close attention to the plot development in the musical. |
| 1 | Affective Experience | I experience emotional resonance while watching the musical. |
| 2 | | I really like the adaptation of the nursery rhymes in the musical. |
| 3 | | I think the adaptation effect of the musical is excellent. |
| 1 | Scene Design | I am satisfied with the stage effects of the musical. |
| 2 | | I think the equipment used in the musical is great. |
| 3 | | I think the scene design of the musical matches the theme. |
| 1 | Audience Interaction | I feel like I can participate in the musical. |
| 2 | | I think the actors interacted effectively with the audience. |
| 3 | | I enjoy the feeling of interaction between the actors and the audience. |
| 4 | | The interaction between the actors and the audience makes me more satisfied. |
| 1 | Audience Satisfaction | I am satisfied with the overall quality of the musical. |
| 2 | | I am satisfied with the performance of the actors. |
| 3 | | I am satisfied with the adaptation of the nursery rhymes in the musical. |
| 1 | Repurchase Intention | I am willing to buy tickets again to watch this musical. |
| 2 | | I am willing to recommend this musical to my friends. |
| 3 | | I may continue to watch this musical in the long term. |

## Supporting information

The data collected from the survey questionnaire is available in the file data (xlsx)

## Author contributions

**Data curation:** Yu Hu.

**Investigation:** Jian Li.

**Methodology:** Jian Li, Yu Hu.

**Validation:** Jian Li.

**Writing – original draft:** Jian Li.

**Writing – review & editing:** Jian Li, Yu Hu.

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
