## [Decision Letter · Decision Letter 0]

13 Jan 2025

PONE-D-24-43964How does the audience return to the music theater? Exploring the Influence of musical theater adaptation of Chinese nursery rhymesPLOS ONE

Dear Dr. Li,

Thank you for submitting your manuscript to PLOS ONE. After careful consideration, we feel that it has merit but does not fully meet PLOS ONE’s publication criteria as it currently stands. Therefore, we invite you to submit a revised version of the manuscript that addresses the points raised during the review process. Both reviewers pointed to issues with the data and its analysis. The article needs to situate better its place in the literature and improve its conclusion, in order to better state its contribution to said literature.

We look forward to receiving your revised manuscript.

Kind regards,

Rafael Galvão de Almeida, PhD.

Academic Editor

PLOS ONE

2. We note that in the online submission form you have mentioned that informed consent was taken, whereas in the manuscript you have stated that the need for consent was waived. Kindly update the Methods and online submission form with the correct information. Also, please mention (1) whether consent was informed and (2) what type you obtained (for instance, written or verbal, and if verbal, how it was documented and witnessed).

3. Thank you for stating the following financial disclosure:  [2021 Featured Innovation Projects of General Colleges and Universities in Guangdong Province"Research on the dissemination mode of dramatized Leizhou Nursery rhymes under the background of Rural Revitalization Strategy" �2021WTSCX271�].  Please state what role the funders took in the study.  If the funders had no role, please state: "The funders had no role in study design, data collection and analysis, decision to publish, or preparation of the manuscript." If this statement is not correct you must amend it as needed.

Additional Editor Comments (if provided):

Reviewers' comments:

Reviewer's Responses to Questions

**Comments to the Author**

1. Is the manuscript technically sound, and do the data support the conclusions?

Reviewer #1: Yes

Reviewer #2: Yes

2. Has the statistical analysis been performed appropriately and rigorously? 

Reviewer #1: Yes

Reviewer #2: Yes

3. Have the authors made all data underlying the findings in their manuscript fully available?

Reviewer #1: Yes

Reviewer #2: Yes

4. Is the manuscript presented in an intelligible fashion and written in standard English?

Reviewer #1: Yes

Reviewer #2: Yes

5. Review Comments to the Author

Reviewer #1: The topic of this manuscript is relatively novel. I hope that it can be successfully published after the seven parts are revised. I strongly recommend uploading the word file and the complete analysis data DOI link next time.

1.Replace Inactive Links:Some external links cited in the article are inactive,especially in the research background and literature review sections,which affects the retrieval of literature.Please check and replace them with active links or directly cite accessible literature to ensure the rigor and traceability of the research.

2.Supplement Research Methods:The article lacks a detailed description of the research methods,making it difficult to understand the specific implementation process and the basis for method selection.It is recommended to organize the research methods separately,covering aspects such as research design,data collection,sample selection,data processing,and analysis,and connect them with the theoretical background or hypothesis section to provide methodological support for hypothesis verification and theoretical discussion.

3.Unify Data Quantity:The actual number of questionnaire data is 311,but the article only mentions 300,which is inconsistent.Please verify the data,and if invalid questionnaires or abnormal data are excluded,clearly state the exclusion criteria and quantity to ensure the accuracy and reliability of the data.

4.Standardize Data Files:The management of data files is not standardized,and the transparency is insufficient.It is suggested to upload the organized SPSS files and the.sav files of the analysis process to open-source platforms such as Mendeley Data and fairsharing.org,and provide the database DOI number for easy access to the data by readers.The questionnaire appendix can be omitted,and the details can be integrated into the data link description.

5.Optimize Assumptions and Models:The logical relationships of the research assumptions are not clear,and the theoretical model does not deeply explore the mediating and moderating effects.It is not enough to simply attach significant coefficients to the assumptions.Please reorganize the assumptions to clarify the causal logic of the variables,such as refining the impact path of audience interaction on emotional experience.

6.In-depth Analysis and Discussion:The result analysis only simply describes the path coefficients and does not deeply interpret the reasons for the non-significant paths and the implications of the results.Please pay attention to the non-significant paths,such as analyzing the reasons for the lack of significant impact of lyrics content depth and artistry on emotional experience from the aspects of lyrics content depth and artistry;if possible,please construct a more comprehensive theoretical model,introducing mediating variables and moderating variables,such as cultural differences and individual differences,to better explain the phenomena and guide practice.

7.Improve Conclusion:The conclusion section is brief and does not fully summarize the research findings and contributions,and the guidance for future research directions is not clear.Please summarize the core findings in detail,such as the key role of audience interaction and emotional experience and their impact mechanisms on satisfaction and repurchase intention;point out the theoretical and practical significance of the research for the development of the musical theater industry,and propose future research questions based on the limitations,such as cross-cultural comparisons and long-term tracking of audience loyalty,etc.

Reviewer #2: The manuscript investigates a timely and underexplored topic: the adaptation of nursery rhymes in Chinese musicals and their impact on audience behavior. The study is well-designed and contributes valuable insights to the field. Below are detailed comments:

1. Abstract: The abstract effectively summarizes the study's objectives, methods, and findings. Consider emphasizing the practical implications for musical production more explicitly.

2. Introduction:

2.1 The introduction provides a strong rationale for the study but could benefit from a clearer articulation of how it fills specific gaps in the existing literature.

2.2 More context on the significance of nursery rhyme adaptations in Chinese musicals would strengthen the argument.

3. Methodology:

3.1 The methodology is detailed and appropriate. The explanation of questionnaire design and the use of validated scales ensure rigor.

3.2 It might be helpful to specify the demographic characteristics of the survey respondents to provide more context for the findings.

4. Results:

4.1 The results are well-presented, with clear tables and figures. However, the discussion of non-significant findings (e.g., the impact of lyrics adaptation on affective reactions) could be expanded to explore possible reasons.

4.2 The inclusion of a visual path model helps clarify the relationships among variables.

5. Discussion:

5.1 The discussion effectively interprets the findings but could be improved by comparing the results more extensively with those of prior studies.

5.2 Consider addressing limitations such as the generalizability of findings beyond the surveyed audience.

6. Practical Implications:

6.1 The recommendations for enhancing audience interaction and satisfaction are practical and grounded in the findings.

6.2 Expanding on how these insights could be applied to different cultural or economic contexts would add value.

6. PLOS authors have the option to publish the peer review history of their article (what does this mean?). If published, this will include your full peer review and any attached files.

Reviewer #1: **Yes: **Weijia Yang

Reviewer #2: No

---

## [Author Response · Author response to Decision Letter 1]

22 Feb 2025

Dear Reviewers,

I hope this message finds you well. I would like to express my sincere gratitude for taking the time to review our manuscript titled "How does the Audience Return to the Music Theater? Exploring the Influence of Musical Theater Adaptation of Chinese Nursery Rhymes." Your insightful comments and constructive suggestions have been invaluable in enhancing the quality of our work. We have carefully considered each of your feedback points and have made comprehensive revisions accordingly.

We believe that these revisions have significantly improved the manuscript, making it more robust, coherent, and impactful. We are confident that the revised version now addresses all the concerns raised and presents a valuable contribution to the field of musical theater studies. Once again, thank you for your invaluable feedback and dedication to improving the quality of our work. We look forward to the next steps in the publication process and hope to see our manuscript published in PLOS ONE.

Warm regards,

Jian Li

ZhanJiang Preschool Education Collage, Zhanjiang, China

---

## [Decision Letter · Decision Letter 1]

10 Mar 2025

How does the audience return to the music theater? Exploring the Influence of musical theater adaptation of Chinese nursery rhymes

PONE-D-24-43964R1

Dear Dr. Li,

We’re pleased to inform you that your manuscript has been judged scientifically suitable for publication and will be formally accepted for publication once it meets all outstanding technical requirements.

Kind regards,

Rafael Galvão de Almeida, PhD.

Academic Editor

PLOS ONE

Additional Editor Comments (optional):

Reviewers' comments:

Reviewer's Responses to Questions

**Comments to the Author**

1. If the authors have adequately addressed your comments raised in a previous round of review and you feel that this manuscript is now acceptable for publication, you may indicate that here to bypass the “Comments to the Author” section, enter your conflict of interest statement in the “Confidential to Editor” section, and submit your "Accept" recommendation.

Reviewer #1: All comments have been addressed

Reviewer #2: All comments have been addressed

2. Is the manuscript technically sound, and do the data support the conclusions?

Reviewer #1: Yes

Reviewer #2: Yes

3. Has the statistical analysis been performed appropriately and rigorously? 

Reviewer #1: Yes

Reviewer #2: Yes

4. Have the authors made all data underlying the findings in their manuscript fully available?

Reviewer #1: Yes

Reviewer #2: Yes

5. Is the manuscript presented in an intelligible fashion and written in standard English?

Reviewer #1: Yes

Reviewer #2: Yes

6. Review Comments to the Author

Reviewer #1: This study focuses on the audience behavior mechanism of Chinese nursery rhyme-adapted musicals. The topic has both academic value and practical significance. The research design is relatively standardized, and the conclusions are of reference value for the practice of the musical industry.

Reviewer #2: (No Response)

7. PLOS authors have the option to publish the peer review history of their article (what does this mean?). If published, this will include your full peer review and any attached files.

Reviewer #1: **Yes: **Weijia Yang

Reviewer #2: No

---

## [Editor Report · Acceptance letter]

PONE-D-24-43964R1

PLOS ONE

Dear Dr. Li,

I'm pleased to inform you that your manuscript has been deemed suitable for publication in PLOS ONE. Congratulations! Your manuscript is now being handed over to our production team.

Kind regards,

on behalf of

Dr. Rafael Galvão de Almeida

Academic Editor

PLOS ONE